# Development of core outcome sets for vision screening and assessment in stroke: a Delphi and consensus study

Fiona J Rowe,[1] Lauren R Hepworth,[1] Jamie J Kirkham[2]

¹Department of Health Services Research, University of Liverpool, Liverpool, UK
²Department of Biostatistics, University of Liverpool, Liverpool, UK

**Correspondence to**
Professor Fiona J Rowe;
rowef@liverpool.ac.uk

## ABSTRACT

**Objectives** Visual impairment following stroke is common with a reported incidence of visual impairment in 60% of stroke survivors. Screening for visual impairment is neither routine nor standardised. This results in a health inequality where some stroke survivors receive comprehensive vision assessment to identify any existent visual problems while others receive no vision assessment leaving them with unmet needs from undiagnosed visual problems. The aim of this study was to define two core outcome sets (COS), one for vision screening and one for full visual assessment of stroke survivors.

**Design** A list of potentially relevant visual assessments was created from a review of the literature. The consensus process consisted of an online 3-round Delphi survey followed by a consensus meeting of the key stakeholders.

**Participants** Stakeholders included orthoptists, occupational therapists, ophthalmologists, stroke survivors and COS users such as researchers, journal editors and guideline developers.

**Setting** University.

**Outcome measures** COS.

**Results** Following the consensus process we recommend the following nine assessments for vision screening: case history, clinical observations of visual signs, visual acuity, eye alignment position, eye movement assessment, visual field assessment, visual neglect assessment, functional vision assessment and reading assessment. We recommend the following 11 assessments for full vision assessment: case history, observations, visual acuity, eye alignment position, eye movement assessment, binocular vision assessment, eye position measurement, visual field assessment, visual neglect assessment, functional vision assessment, reading assessment and quality of life questionnaires.

**Conclusions** COS are defined for vision screening and full vision assessment for stroke survivors. There is potential for their use in reducing heterogeneity in routine clinical practice and for improving standardisation and accuracy of vision assessment. Future research is required to evaluate the use of these COS and for further exploration of core outcome measures.

## INTRODUCTION

Visual impairment is common in stroke occurring in up to 73% of stroke survivors.[1] Visual impairment is typically categorised into impairments of central vision, eye movements, visual fields and visual perception.[2] Vision is arguably our most important sense. Visual impairment results in impaired activities of daily living with reduced quality of life through loss of independence, greater risk of trips and falls and accidents.[3–6] This leads to loss of independence and potentially results in social isolation and depression.[5 6]

The primary focus of stroke rehabilitation is often occupational therapy and physiotherapy to mobilise patients, improve limb function and balance and engage in activities of daily living plus speech and language therapy for communication difficulties.[7] Many rehabilitation strategies require visual input, for example to safely mobilise around potential obstacles, recognise depth and position of objects and recognise visual cue cards. Given so many stroke survivors have visual impairment it is important to screen for this at an early time point post-stroke onset with the aim to optimise the rehabilitation process.

The recent Impact of Visual Impairment after Stroke (IVIS) study reported specialist orthoptist vision screening is possible at a median of 3 days post-stroke onset with the majority of stroke survivors being assessed within 1 week of stroke onset.[1] This study used standardised visual assessment methods with portable equipment to be used at the

## Strengths and limitations of this study

► This study followed robust methodologies in accordance with Core Outcome Measures in Effectiveness Trials-initiative guidelines.
► Two core outcome sets are produced for vision screening and for full assessment of stroke survivors with visual impairment.
► There is potential for their use with other types of acquired brain injury causing visual impairment.
► Attrition rates were moderate but similar to other Delphi surveys.
► Larger response numbers, including international participants, would be of benefit.

patient's bedside. There is, however, no standardised visual screening assessment for post-stroke visual impairment. In one UK survey it was found that 45% of stroke services provided no formal vision assessment for stroke patients.[8] A further survey of practice identified that only 7% of stroke units had a policy relating to vision assessment and management.[9] Both surveys showed lack of standardisation for vision assessment and treatment for stroke survivors. The National Stroke Strategy argues that vision and visual perceptual difficulties are components requiring multifaceted stroke specific rehabilitation and support.[10] The Royal College of Physicians recommend that every patient with stroke has a practical assessment of vision and examination of the visual field.[11]

On the basis that there is no consensus on how to adequately screen for visual impairment after brain injury, the aim of this study was to achieve consensus on the content of vision screening and full vision assessment for stroke survivors in order to better identify visual impairment. Screening and/or full vision assessments are to be undertaken at any time point post-stroke onset with the intention that identification of visual impairment enables prompt access to earlier visual rehabilitation options. One approved process to reach consensus on screening and assessment for specific conditions is through the development of core outcome sets (COS).[12] COS indicate the minimum that should be measured and reported in all studies of a specific condition. The overall purpose of a vision screening and full assessment COS is to improve routine care in the UK National Health Service (NHS) through standardisation of assessments.

In this study we report the results of a Delphi process and consensus meetings in the development of a COS for vision screening and a COS for full vision assessment of stroke survivors. Vision screening was defined as the assessments considered important for use by clinicians not working in eye care settings and without formal experience or training in performing eye tests. Full vision assessment was defined as the assessments considered important for use by clinicians who had formal eye care training and were principally based in eye clinics and providing more detailed assessment than possible with screening assessment of basic levels of visual function.

## METHODS
### Ethical approval
Informed consent was obtained for the Delphi survey—the participant checked a consent tick box on the opening page of the survey. Informed written consent was obtained from all participants in the consensus meeting.

Development of the COS involved three phases: (1) the generation of a comprehensive list of outcomes; (2) a Delphi survey involving three rounds to gain consensus as to which outcomes are most important; and (3) patient and professional consensus meetings to agree a final COS. A protocol for the development of this COS project was written by the steering committee, registered in the Core

Outcome Measures in Effectiveness Trials (COMET) initiative website (http://www.comet-initiative.org/studies/details/275?result=true) and available as open access (http://pcwww.liv.ac.uk/~rowef/index_files/Page356.html). When developing this COS, we followed the minimum set of development standards set out by Core Outcome Set-STAndards for Development Statement, and report the results against the Core Outcome Set–STAndards for Reporting guideline.[13 14]

### Steering group
A steering group was set up to inform the development of the various stages of this study and to discuss the results at each phase of the study. It comprised a representative from clinical professions involved in the vision screening of patients with acute stroke brain injury (orthoptist, occupational therapist and neuro-ophthalmologist), two stroke survivors and COS users (clinical and academic researchers, systematic reviewer and clinical guideline developer); eight members in total. Clinicians were identified from a national orthoptist stroke clinical advisory group, a regional allied health profession research network and regional ophthalmology research network. Stroke survivors were identified from a national stroke/vision patient and public involvement panel. COS users were identified from local university staff with links to the COMET initiative and/or those involved in national clinical guideline panels. The COMET initiative brings together people interested in the development and application of agreed standardised sets of outcomes (http://www.comet-initiative.org/).

### Patient and public involvement
The development of this research question was informed by patients' priorities, experience and preferences outlined by a National Institute for Health Research James Lind Alliance priority setting partnership. Patients were involved in the design of this study as members of the steering group set up specifically for the study and outlined previously. Further, patients were involved in the recruitment to and conduct of the study through the VISable patient and public involvement panel which is described below in the next section. Results will be disseminated to study participants by emailing a copy of the full publication of this research along with a lay summary of the results.

### Stakeholders
For participation in this study, we sought the input of clinicians (eligibility=those involved in the vision screening of patients with acute brain injury), patients (eligibility=those with brain injury causing visual impairment), caregivers (eligibility=those caring for people with brain injury causing visual impairment), editors of journals and COS users. Clinicians, journal editors and COS users were invited to participate through advertisements circulated via national professional societies (eg, British and Irish Orthoptic Society, Royal College of Occupational

Therapists), via international orthoptic editorial groups (eg, *Strabismus* and *British and Irish Orthoptic Journal*) and via regional allied health profession research networks (Council for Allied Health Professions Research). Stroke survivors and caregivers were invited through advertisements circulated via a national stroke/vision patient and public involvement panel (VISable) and through national charities (eg, the Stroke Association, Royal National Institute for the Blind).

## Phase 1
### Literature review

In order to develop a preliminary list of outcomes for a Delphi survey we undertook an overview of seven systematic reviews of studies/trials reporting vision screening, assessment and treatment of post-stroke visual impairment.[2 15–20] We extracted 119 outcomes, many of which were variations on test choices for specific visual functions. Therefore these were combined into one outcome domain (table 1).

COS do not address how tests should be defined or measured. Thus this combination decision was made by the steering group who decided through discussion that test choice would vary on a case-by-case basis dependent on the patient's ability to undertake the test and that COS should consider only the general assessment needed and not the specific methods of measurement. For example, logMAR, Snellen, fixation/following, grating cards and fixation distance were outcomes combined under one outcome domain of 'visual acuity' assessment.

This process produced a list of 22 domains for vision screening and a list of 24 domains for full visual assessment. These lists were then circulated to the VISable patient and public involvement panel for approval and checking of writing for lay terms as the basis of the online survey development. VISable advised that patient-important outcomes were covered by the outcome domains and did not add any further outcomes to those already identified.

## Phase 2
### Delphi survey

We undertook a prospective consensus study using a Delphi process. Delphi is a structured process that is widely used in developing core outcomes sets.[21] The process aims to achieve consensus through the collection of stakeholder opinions.

SurveyMonkey (SurveyMonkey, 2015) was used as the online platform to administer the Delphi process. The survey was piloted for checks of ambiguity and appropriate use of lay language by the steering group and then released live.

An email outlining the project and survey was sent out via stakeholder networks (as outlined earlier in the stakeholder section) with a snowballing technique of onward roll-out from individual stakeholders. The Delphi survey consisted of three rounds (figure 1). Rounds remained open for 10 weeks, with regular 2-week reminders sent to those that had partially completed or not completed to maximise response rates.

In round 1, the Delphi started with an introductory page, which outlined the purpose of the study and how to complete each section, as follows:

"We are not asking you to provide detailed answers to these items, but to judge how important they are when assessing visual impairment in brain injury. Please view the items listed below and score their importance with 1 being not important and 9 being critical. After a brain injury, patients may first have visual screening to detect if a visual problem is present. Once a visual problem is found, patients may then receive a full visual assessment. In part 1a, please rate the items below as to how important you believe they are to a *screening assessment*. In part 1b please rate the items below as to how important you believe they are to a *full visual assessment*."

All terms had explanatory notes to aid interpretation. All participants were asked to score each assessment in terms of importance and asked to identify any additional outcomes of importance that did not appear in the list of assessments. In the acknowledgement that definition of importance varies between individuals and between stakeholder groups, the Delphi survey was circulated widely to a variety of stakeholder groups to obtain greater consistency in responses. Additional outcomes added in round 1 were reviewed by the steering group to consider their relevance to the survey and to identify and remove duplicates.

In round 2, all data from round 1 were analysed and compiled by stakeholder groups; (1) stroke survivors/carers, (2) stroke team professionals and (3) eye team professionals to allow different perspectives to be considered prior to re-rating.[12] The summarised results for each stakeholder group were provided to all participants who fully completed the survey in the first round. Each participant was also shown their personal original score for each domain. Participants had the opportunity to re-score based on the summary scores from the three stakeholder groups versus their previous personal scores. All participants were also asked to score on additional outcomes that were identified and added in round 1.

In round 3, the results from round 2 were analysed. Following analysis, the results from each stakeholder group were reviewed by the steering group and judged to be similar in terms of percentage spread across the responses of 1–9. Thus they were presented as one compilation of all responses. Each participant was shown their personal score from round 2 for each domain along with the summary scores from all other participants, and asked to score each domain again in terms of importance.

Scoring method: In each round, participants were asked to score the importance of each domain listed on a 9-point scale (1–3: not important; 4–6: important but not critical; 7–9: critical, as well as an 'unable to score' option). The scale was devised by the Grading of Recommendations Assessment, Development and Evaluation (GRADE) working group[22] to score the quality of

| Table 1 | Outcome extraction | |
| --- | --- | --- |
| **Screening assessment** ⬅ | **Outcomes extracted from reviews** | ➡ **Full assessment** |
| Case history—open questions | Case history—open questions | Case history—open questions |
| Case history—specific questions | Case history—specific questions<br>► Eye strain<br>► Reading difficulty<br>► Blurred, altered or reduced vision<br>► Visual field loss<br>► Awareness of full environment<br>► Oscillopsia<br>► Diplopia<br>► Polyopia<br>► Visual hallucinations<br>► Altered colour vision<br>► Altered movement of objects<br>► Depth perception misjudgements<br>► Tilted images<br>► Distorted images<br>► Face/object recognition<br>► Clutter difficulty<br>► Getting lost<br>► Prolongation of images<br>► Reverse image size<br>► Glare<br>► Visual crowding<br>► Visual disorientation | Case history—specific questions |
| Case history—carer open questions | Case history—carer open questions | Case history – carer open questions |
| Case history—carer specific questions | Case history—carer specific questions<br>► Personal care issues<br>► Eyes constantly moving/jerking<br>► Missing things to one side<br>► Bumping into things<br>► Concerns over vision<br>► Visual hallucinations<br>► Family/friend recognition<br>► Difficulty naming objects<br>► Getting lost<br>► Reading problems | Case history—carer specific questions |
| Case history—previous ocular history | Case history—previous ocular history | Case history—previous ocular history |
| Case history—glasses wear | Case history—glasses wear | Case history—glasses wear |
| Observations—open comments | Observations—open comments | Observations—open comments |
| Observations—specific features | Observations—specific features<br>► Lids<br>► Pupils<br>► Squint—misaligned eyes<br>► Eye movements<br>► Turning head to see<br>► Closing one eye to see better<br>► Misjudging distances<br>► Wobbling eyes | Observations—specific features |
| Visual acuity | logMAR charts<br>Snellen charts<br>Fixation and following observation<br>Vanishing optotype charts<br>Grating charts<br>Near acuity charts<br>Kay's pictures<br>Sheridan Gardiner single optotypes<br>Lea symbols<br>Fundus check<br>Retinal photography/OCT | Visual acuity |
| Eye alignment position | Cover uncover test<br>Alternating cover test<br>Observations of corneal reflections | Eye alignment position |

 Rowe FJ, *et al. BMJ Open* 2019;**9**:e029578. doi:10.1136/bmjopen-2019-029578

**Table 1** Continued

| Screening assessment ← | Outcomes extracted from reviews | → Full assessment |
|---|---|---|
| Eye movement assessment | Nine positions of gaze<br>Horizontal gaze only<br>Vertical gaze only<br>Horizontal and vertical gaze only<br>Vergence<br>Saccade movement<br>Smooth pursuit movement<br>Optokinetic nystagmus movement<br>Vestibulo-ocular reflex<br>Hess/Lees/Harms wall charts | Eye movement assessment |
| Binocular vision assessment | Retinal correspondence<br>Sensory fusion<br>Motor fusion<br>Stereopsis | Binocular vision assessment |
| Eye alignment measurement | Prism cover test<br>Krimsky test<br>Prism reflection test<br>Synoptophore<br>Bruckner test<br>Maddox rod | Eye alignment measurement |
| Visual field assessment | Confrontation<br>Static central perimetry<br>Static peripheral perimetry<br>Kinetic perimetry | Visual field assessment |
| Visual neglect assessment | Line bisection<br>Cancellation task—star, balloon, heart, etc.<br>Clock drawing<br>Room/environment description<br>Behaviour inattention test battery | Visual neglect assessment |
| Functional assessment | Observed navigation<br>Reading<br>Eye scanning<br>Walking observations<br>Activities of daily living<br>Self-care<br>Body placement<br>Spatial awareness<br>Mobility observations<br>Writing<br>Hand–eye coordination<br>Visual memory and cognition<br>Visual perception | Functional assessment |
| Reading assessment | Special test, eg Wilkins, iReST, Radner Newspaper, magazine, book | Reading assessment |
| Questionnaires | Vision-related, eg VFQ25, DLDV<br>Health-related, eg SF12<br>Activity of daily living, eg IADL<br>Extended activity of daily living, eg NEADL | Questionnaires |
| Pupil assessment | Swinging flashlight test | Pupil assessment |
| Lid assessment | Palpebral apertures<br>Lid function test | Lid assessment |
| Contrast sensitivity assessment | Pelli-Robson chart<br>Mars test<br>VisTech | Contrast sensitivity assessment |
| Colour vision assessment | Ishihara test<br>City test | Colour vision assessment |

DLDV, Daily Living tasks Dependent on Vision; IADL, Instrumental Activities of Daily Living; iReST, International Reading Speed Test; NEADL, Nottingham Extended Activities of Daily Living; OCT, Optical Coherence Tomography; SF12, Short Form 12; VFQ25, Visual Function Questionnaire 25.

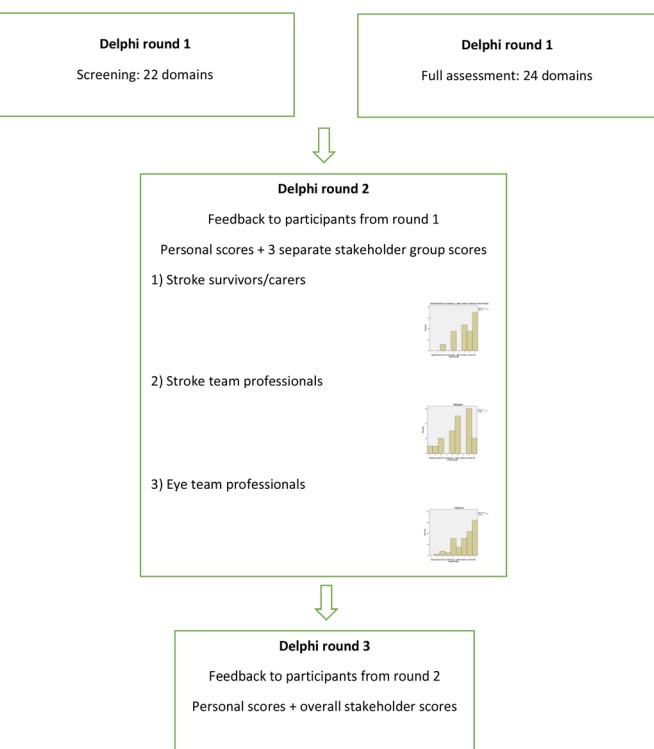

**Figure 1** Flow chart of Delphi process across three survey rounds.

evidence for outcomes in systematic reviews and has been adopted in other COS development work research using Delphi methods.

Methods of analysis: For each round of Delphi, descriptive statistics were used to summarise the results for each domain, including the percentage of participants scoring the domain at each possible response from 1 to 9.

Consensus was defined a priori; however, this information was not provided on the Delphi survey. Participants were aware of these cut-off values at the time of attending the consensus meeting: 'Consensus in' (ie, consensus that the domain should be included in the core set) was defined as greater that 70% of participants scoring as 7–9 and less than 15% of participants scoring 1–3. 'Consensus out' (ie, consensus that the domain should not be included in the core set) was be defined as greater than 70% of participants scoring as 1–3 and less than 15% of participants scoring as 7–9. All other combinations were seen as equivocal. The domains that were designated as 'consensus in' or seen as 'equivocal' were taken forward and discussed in more detail at the consensus meeting for inclusion into the final COS, one COS for screening and one COS for full assessment. Participants were reminded of all domains not reaching consensus as part of the Delphi process.

### Phase 3
#### Consensus meeting
Representatives from the Delphi survey were invited to a face-to-face consensus meeting. All round 3 survey completers (n=51) were emailed an invitation. All

stakeholders were reasonably represented. The format of the meeting included a short study overview, a presentation containing a summary of the results and number of domains reaching consensus from the survey. Each consensus domain was considered in turn, in order of their presentation in the Delphi survey, to ratify these results. Each remaining domain was then considered, in turn, with full discussion. Similar to the survey, each domain was considered as reaching consensus with 70% of participants voting in favour of its inclusion.

Each participant voted on every domain being asked to vote 'Yes' (this domain should be included in the COS), 'No' (this domain should not be included) or 'Unsure' using voting slips. After voting for all domains was completed, the results were collated during a rest break for the participants. When the meeting resumed, the results were presented to the group. Domains were retained or dropped when consensus was reached. Discussion and further rounds of voting, restricting the options to 'Yes' or 'No', were undertaken until consensus was reached on all domains. All domains retained were included in the final COS.

### RESULTS
Figure 2 outlines a flowchart of the results for number of participants and number of domains.

### Phase 1
#### Outcome identification
An overview of systematic reviews produced a list of 119 outcomes. Outcomes that were variations on test choices for specific visual functions were combined into outcome domains. This produced 22 domains for vision screening and a list of 24 domains for full visual assessment.

### Phase 2
#### Delphi survey
In total 123 participants registered for round 1 of the Delphi survey. Six did not complete the survey. The remaining participants comprised 79 orthoptists, 20 occupational therapists, 17 stroke survivors and 1 ophthalmologist. There were 20 males and 97 females with ages ranging from 18 to 84 years (figure 3). In round 2, 65 participants completed the survey—an attrition of 44.5% from round 1. These participants comprised 47 orthoptists, 10 occupational therapists, 7 stroke survivors and 1 ophthalmologist. In round 3, 51 participants completed the survey—an attrition of 56.4% from round 1 (78.5% from round 2). These participants comprised 39 orthoptists, 6 occupational therapists, 5 stroke survivors and 1 ophthalmologist.

Following completion of the survey of 46 domains in round 1, 12 additional outcomes were put forward by the participants for inclusion in round 2. These comprised five screening outcomes and seven full assessment outcomes. After round 2 of 58 domains, three outcomes were excluded; these were test variations

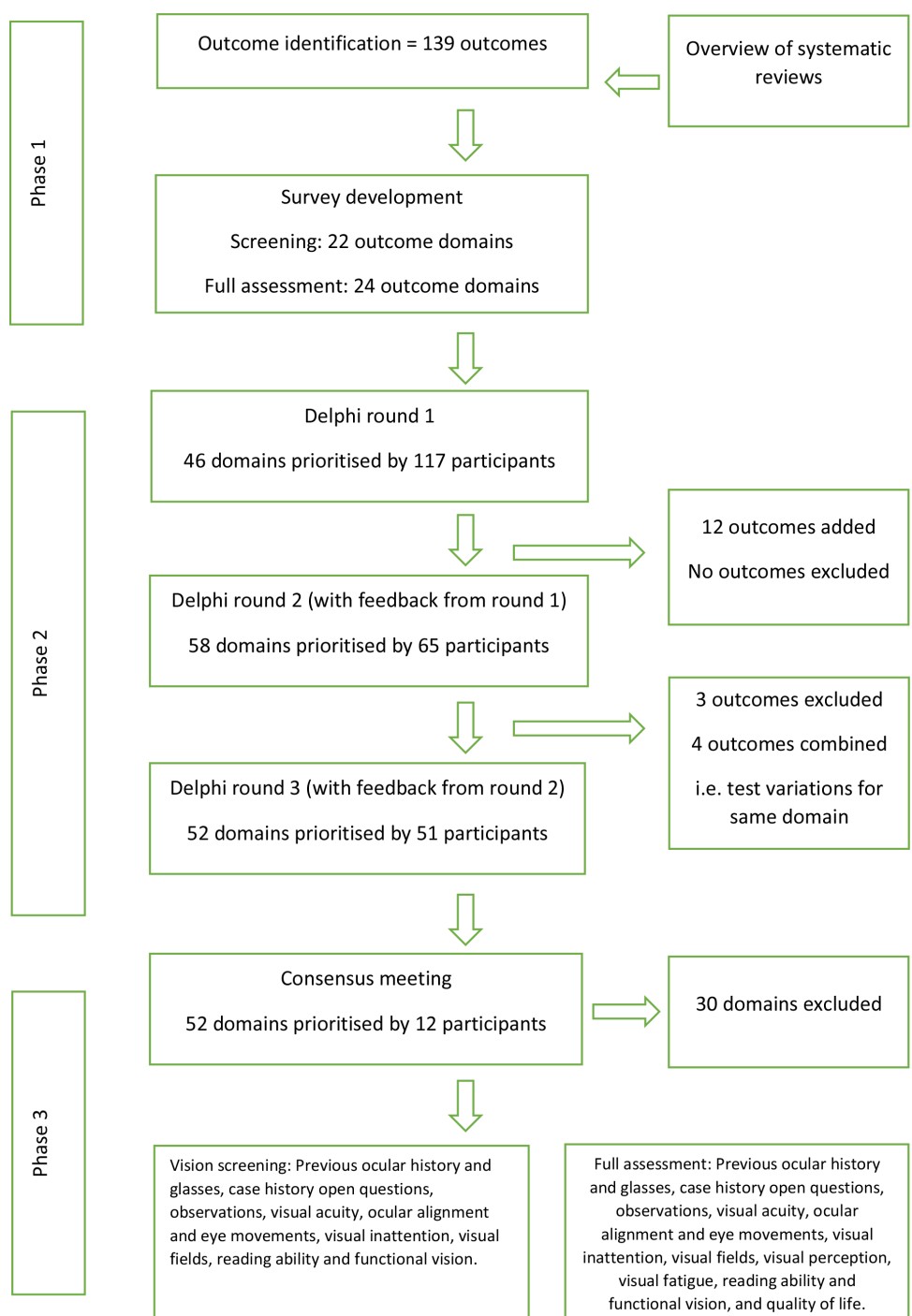

**Figure 2**  Flow chart of consensus process across three phases of outcome identification, Delphi survey and consensus meeting.

already included within the existent domains. Four outcomes were combined; these were variations on test choices and could be combined into one domain—as per the process outlined in phase 1. No new outcomes were introduced. Following round 3 of 52 domains, consensus was achieved for 14 screening domains leaving ten domains for discussion, and consensus was achieved for 22 full assessment domains leaving six domains for further discussion.

### Phase 3
#### Consensus meeting

The consensus meeting was a 1 day event held in Liverpool, UK with 12 participants comprising five occupational therapists (2 with research roles), 3 orthoptists (1 with a research role), 2 patients, 1 Cochrane editor and 1 facilitator. The facilitator did not take part in the voting. The objective of the meeting was to discuss and vote on the Delphi domains—the results of the Delphi survey had

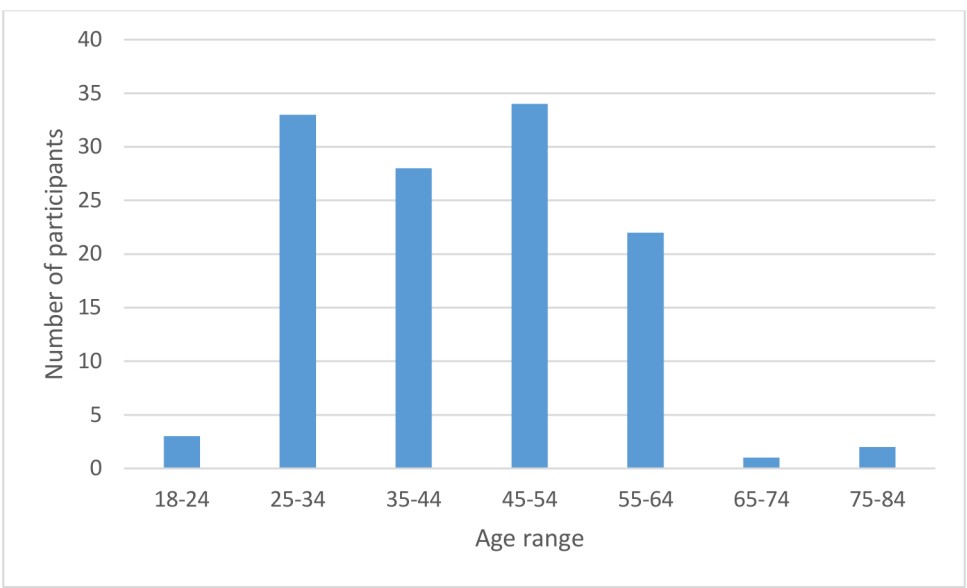

**Figure 3** Age range of participants.

been provided to participants prior to and during the meeting.

*Screening*

Online supplementary table 1 shows the ranking for general tests considered for vision screening. The Delphi survey had reached the following consensus:

► Full consensus for inclusion was obtained for case history asking patients/carers open and specific questions plus checks of previous ocular history and glasses wear. Full consensus was obtained for observations, tests of visual acuity, eye alignment, eye movements, visual fields, visual inattention and functional visual assessment.

► Full consensus for exclusions included contrast sensitivity and colour vision assessments.

► No consensus was reached for command saccades, binocular vision assessment, alignment measurement, reading assessment, use of questionnaires, pupil and lids assessment and screening for visual memory and recognition.

During the consensus meeting, final consensus for vision screening included nine domains of previous ocular history and glasses, case history open questions, observations, visual acuity, ocular alignment and eye movements, visual inattention, visual fields, reading ability and functional vision (Box 1). Experts estimated a vision screening assessment of no more than 20 min with these domains based on the use of basic screening tests of visual functions.

*Full vision assessment*

Online supplementary table 2 shows the ranking for general domains considered for full vision assessment. The Delphi survey had reached the following consensus:

► Full consensus for inclusion was obtained for case history asking patients/carers open and specific questions plus checks of previous ocular history and glasses

wear. Full consensus for observations, tests of visual acuity, eye alignment, eye movements, binocular vision, alignment measurement, visual fields, visual inattention, reading, lids and pupils, visual perception and functional visual assessment.

► Full consensus for exclusions included retinal photography and OCT.

---

**Box 1    Core outcome sets**

**A Vision screening core outcome set**
**Vision screening COS**
► Case history—previous ocular history and use of glasses, open questions
► Observations
► Visual acuity
► Eye alignment position
► Eye movement—ocular motility assessment
► Visual field assessment
► Visual neglect assessment
► Functional vision assessment
► Reading assessment

**B        Vision full assessment core outcome set**
**Full vision assessment COS**
► Case history—previous ocular history and use of glasses, open questions, visual fatigue and visual perception questions
► Observations—including lids and pupils
► Visual acuity
► Eye alignment position
► Eye movement—ocular motility assessment
► Binocular vision assessment
► Eye position measurement
► Visual field assessment
► Visual neglect assessment
► Functional vision assessment
► Reading assessment
► Quality of life questionnaires

► No consensus was reached for eye contact during conversations, use of questionnaires, contrast sensitivity, colour vision and fundus checks.

During the consensus meeting, final consensus for full vision assessment included 12 domains of previous ocular history and glasses, case history open questions including visual fatigue and visual perception questions, observations including lids and pupils, visual acuity, ocular alignment and eye movements, binocular vision, visual inattention, visual fields, reading ability and functional vision, and quality of life measurement (box 1). These comprise detailed assessments of visual functions using tests validated as reliable and repeatable.

## DISCUSSION

A COS is an agreed minimum set of outcome measures that should be reported. By reporting a minimum set of measures, this reduces the heterogeneity of outcomes across studies, which, in turn, supports future evidence synthesis.[12–14] Once a COS is defined, it is important to achieve consensus on how the outcomes should be measured.[23]

Delphi and consensus process methods have been used extensively for research-oriented COS but has also been reported in the development of a test battery for visual perception screening[24] and Traumatic Brain Injury (TBI)-related visual impairment.[25] To our knowledge this is the first study to use these methods to develop test batteries for general vision screening and assessment for post-stroke visual impairment.

The goal of this study was to agree a COS to be used in the vision screening and full assessment of visual impairment following stroke. After round 3 and the consensus meeting, stakeholders agreed with the proposed test battery for vision screening consisting of 9 domains and for full vision assessment consisting of 11 domains. The vision screening COS is intended for clinicians who have no formal training or experience in providing eye tests. However, it is acknowledged by expert consensus that the specific sections of visual inattention and functional vision assessments are key areas where assessments by members of the stroke team (eg, by occupational therapists) do not require formal eye training. The full vision assessment COS is intended for specialist vision assessment by professions such as orthoptists and ophthalmologists. Of importance, these COS are the core identified assessments to be made wherever possible when screening or fully assessing the visual functions of stroke survivors. These COS do not exclude further visual assessments which should be added as appropriate and relevant for the individual stroke survivor. Equally, some COS measures may not be achievable due to patient cognitive, communication and physical impairments which prevent assessment.

There are a number of strengths for this study. Two COS have been produced, one for vision screening and one for full vision assessment which addresses a gap in evidence-based practice for post-stroke visual impairment.

Each COS is composed of domains which were ranked by Delphi and consensus opinion from multidisciplinary experts and stroke survivors. As these COS are derived from clinical and research stakeholders, and patients and carers, they are potentially suitable for research in addition to clinical practice. Furthermore they are based on existent outcomes that are easily accessible for implementation, and accepted as validated clinical measures. Although these COS were developed for stroke, there is potential for their use with other types of acquired brain injury causing visual impairment.

There are limitations to this study. Participants were from UK and Ireland and largely represented views and practices within the NHS. Thus, assessments are those used clinically in these countries. It would be valuable to seek expert opinion from international participants regarding the key assessments and their specific outcome measures. However, we recognise that the assessment domains selected for the COS are those that are widely used internationally in clinical and research settings as evidenced from our initial literature review.[2 19 21] We had a limited response by ophthalmologists in the Delphi process, with none subsequently involved in the consensus meeting. However, within the UK and Ireland, orthoptists are the clinical professionals predominantly responsible for the diagnosis and management of stroke-related visual impairment. Thus the greater response from orthoptists in the Delphi survey and consensus process was expected and, we believe, provided robust information required for this study. In other countries, decisions should be taken as to whether involvement of other clinical professionals in the care of these patients would alter the COS lists. We also experienced attrition bias across the Delphi rounds. The attrition rate at round 2 was 44.5% and, by round 3, was 56.4% from the initial sample. High attrition rates however are common using Delphi methods and our rates can be considered within an acceptable range based on those reported for other COS developments.[26] Nevertheless a larger sample would have been preferable. Researcher bias is also a potential limitation. We aimed to limit this by providing a summary of results across all rounds of the Delphi survey and with final decisions left to the consensus meeting with experts. There can be risks from using Delphi methods in which participants can have very disparate views of each outcome. However, we sought a wide variety of participants across a number of stakeholder groups to achieve greater consistency in responses and balance potential outlier responses. Further, this core outcome development process included a final stage of consensus meeting such that decisions were not purely made from the Delphi responses.

## CONCLUSIONS

This study reports the use of Delphi and consensus methods in the development of COS for vision screening and full vision assessment of stroke survivors. Vision screening comprises 9 assessment domains and full vision

assessment comprises 11 domains. These COS will facilitate standardisation of screening and assessment of post-stroke visual impairment while also having the potential to reduce heterogeneity in assessment in future research. Further research is now required to evaluate the use of these COS and outcome measures.

**Acknowledgements** We thank all participants involved in the Delphi survey and consensus meeting. We also thank our clinical and patient colleagues who contributed to the study steering group.

**Contributors** FJR provided oversight for the study and led the writing of the paper. FJR, LRH and JJK contributed to data collection, reviewing the draft paper and approving the final version.

**Funding** This article/paper/report presents independent research funded by the National Institute for Health Research (NIHR: CDF-2012-05-126). The views expressed are those of the authors and not necessarily those of the UK National Health Service, the NIHR or the Department of Health.

**Competing interests** None declared.

**Patient consent for publication** Not required.

**Ethics approval** This study had institutional ethical approval (Ref-1415-040) and was undertaken in accordance with the Tenets of Helsinki.

**Provenance and peer review** Not commissioned; externally peer reviewed.

**Data availability statement** Data are available upon reasonable request.

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
