## [Reviewer comments · BMJ Open]

ARTICLE DETAILS

TITLE (PROVISIONAL)	Development of core outcome sets for vision screening and assessment in stroke: a Delphi and consensus study.
AUTHORS	Rowe, Fiona; Hepworth, Lauren; Kirkham, Jamie

VERSION 1 – REVIEW

REVIEWER	Varshini Varadaraj Johns Hopkins Wilmer Eye Institute
REVIEW RETURNED	11-Mar-2019

GENERAL COMMENTS	Authors present a well written paper that nicely describes the development of core outcome sets for vision screening (by non-eyecare personnel) and full visual assessment (by eyecare professionals) of stroke survivors using a Delphi survey. I commend the authors on a nicely written paper and for employing appropriate and rigorous methodology to report the core outcome sets. Please see minor comments below- Page 3, line 55- What is “observation”? Page 4, line 23- Provide full form for the abbreviation IVIS Page 5, line 5- Provide full form for the abbreviation NHS It is surprising that only 1 ophthalmologist was involved in the initial phases of the Delphi process and no ophthalmologist was present at the consensus meeting. I would think that it is important that ophthalmologists were better represented given that they are the primary eyecare personnel to diagnose and treat visual impairment in stroke survivors. Please acknowledge this in the limitations. In the abstract and main text, instead of re-listing each of the 9 visual assessments from the screening for the full vision assessment, it will be easier for the reader if the authors say- 9 assessments + Binocular vision assessment + Quality of life questionnaires. It is unclear why non-eyecare personnel cannot also perform the binocular vision assessment and quality of life questionnaires if they are able to perform the 9 other tests. It is also unclear how much more information these 2 additional tests will actually provide in helping to treat vision problems in stroke survivors.
---

REVIEWER	Kimberly Patrice Hreha University of Texas Medical Branch, USA Assistant Professor
-----------------	--

REVIEW RETURNED	18-Mar-2019
-------------

GENERAL COMMENTS	I would like to start by saying that I have read many articles by Dr. Rowe and have much respect for her work. I think this article should be published and do not have comments regarding the methodology, grammar, scientific rigor, etc. I do have to mention one point, which relates to this article as well as is just a general comment/suggestion . The term "visual neglect/inattention". I would politely like to make a point to this terminology because I think this word confuses the reader. "Visual neglect/inattention" is the same thing so why put both words? Before my dissertation defense, I looked up the definition of "neglect" and then "inattention" and both are defined: "to pay little or no attention to". I think using multiple words to define this disorder is a HUGE problem. I advocate for the word "spatial neglect" as it is all encompassing. Then after stating that the person has spatial neglect then talk about the type of spatial neglect that the person has. So in this case, I would say that it is affecting the "visual domain". So many researchers and clinicians use different terminology to describe this disorder and this is one reason why I think spatial neglect is under-detected. Also, I have to comment that in this article, 2 of the outcomes that found from the reviews, for "assessing visual neglect" was room environment description and clock drawing. I would suggest these are measuring a different type of spatial neglect. I.e: the room description (from memory) tests representational neglect rather than spatial neglect of the vision domain. Seminal article: https://www.ncbi.nlm.nih.gov/pubmed/16295118/ Also, is the clock drawing also an assessment the visual domain? See here: https://www.ncbi.nlm.nih.gov/pmc/articles/PMC3371137/
--

REVIEWER	Ania Busza University of Rochester School of Medicine and Dentistry USA
-----------------	---

REVIEW RETURNED	12-Apr-2019
-------------

GENERAL COMMENTS	SUMMARY This study uses Delphi and consensus methods to identify core outcome sets for visual screening and visual assessment in patients who have recently suffered a stroke. Visual impairment is common after stroke, and can have significant impact on patient quality of life, yet is often overlooked during the stroke evaluation and treatment. The authors aim to increase post-stroke visual screening and create a standardized set of outcome measures used for visual screening and assessment. In this paper, they describe the methods they use to identify core outcome sets for visual screening (assessments done by clinicians without formal eye/vision testing experience) and for full visual assessments (assessments performed by clinicians with formal eye training). The paper is well written and is a good first step towards addressing an area of stroke-related impairment which has traditionally been under-represented in stroke recovery research despite its important effects on patient quality of life. While the Delphi method / consensus methods can have limitations, it is not unreasonable to use such methods to facilitate discussion between stake holders (in this case a combination of practitioners, stroke survivors and researchers) to identify key elements that should be included in a core outcome set. Creating such core outcome sets is
---

a critical first step for future research studies evaluating how best to identify and treat patients with visual impairments.

CONCERNS WITH REGARDS TO METHODS:

1. Page 7, line 18 – The authors state that “combination decisions” were made by the steering group. My understanding of a “combination decision” is that it a term used to describe when an individual makes a decision after listening/integrating views of others. Did one person of the steering group make the decision or was it achieved through another mechanism? And why is “decisions” plural?

2. I was surprised that no neurologists/stroke advanced practitioners were involved in the survey. This may be due to variations in medical systems between different countries. Where I practice the doctor or advanced practitioner is often the only person who screens the patients for deficits (and then requests help from occupational therapists or eye experts) – therefore their input, at least in identifying best screening outcome measures, may be of interest.

3. What was the logic behind separating results in phase 2 round 2 (page 8, 3rd paragraph) into 3 different stake holder group outcomes? This could lead to individual groups reinforcing their biases, is that part of the intent and if so how is that helpful for this method? Also, given that the steering group changed this strategy and provided one compilation of all responses in round 3 after judging the stakeholder group results to be “similar in terms of percentage spread across the responses of 1 to 9” (p8, lines 34-35), will the graphs be made available as supplementary material, so that the reader can assess the differences/percentage spread themselves?

CONCERNS WITH REGARDS TO CONTENT/CONCLUSIONS:

Some of the caveats with regards to using such a consensus process were not sufficiently addressed. Specifically:

1. Participants were asked to score the “importance” of each domain in the Delphi survey. I do not have access to what instructions they were given, but it seems to me that it would be important to pre-define the ultimate goal prior to starting such a survey. What makes a domain “important”? Is it the impact on patient quality of life? Is it the impact on patient ability to participate in rehabilitation for other stroke-related deficits? Financial considerations, or relative benefit-for-time-unit-spent doing the assessment / cost-effectiveness? It may be that no further instructions were given to the survey participants, but the fact that the determination of “importance” may vary depending on the survey participant’s values should be noted in the discussion.

2. Similarly, were participants aware of the planned Consensus in/out definitions (as described in p8 lines20) when they were scoring the outcome measures? (and were these definitions pre-defined or determined after reviewing the results?)

3. The authors correctly point out that the low sample number and high attrition rates are limitations to the study, but they should also include some critical comments about the risks of using the Delphi method. For example, if a majority of those sampled have a misconception about one of the items evaluated, this type of group consensus will only further perpetuate/reinforce the misconception.

	OTHER DETAILS:  1. JK is listed as an author but has no contributions specified (p2, lines3-7) 2. Page 7, line 9 - Reference 15 refers to the Delphi method itself and therefore probably was not intended to be referred to as one of the 7 systematic reviews used to compile a preliminary list of outcomes 3. Page 8, lines 18-30 – the methods of round 2 were confusing to me. I ultimately concluded that this meant that each participant who completed round 1 received 2 results back to them, their original score AND the summary of the scores from their corresponding group. If this interpretation is correct, perhaps the sentence “the results were presented by these stakeholder groups” (line 22-23) could be rephrased...I initially interpreted this as “the stake holder groups” gave a presentation to the clinicians patients (as opposed to this describing a way of sorting/organizing and then presenting data) 4. Page 8, line 49 – there appears to be an “of” missing in “to score the quality [of?] evidence for outcomes” 5. The link to the COMET initiative website (http://http://www.comet-initiative.org/studies/details/275?result=true) did not work for me when I attempted to access it. I would recommend that the authors make the protocol available as supplementary material for readers. 6. I was confused by the numbers of domains during different phases of the Delphi survey. Specifically after Phase 2 (p10, lines 47-58) the authors state that 3 outcomes were excluded and that “four outcomes were combined” and “could be combined into one domain” (line 54-55). I interpret that as 4 outcomes were combined into 1, which means that 3 would be subtracted. Therefore a total of 6 outcomes were removed between Phase 2 and Phase 3 – yet in figure 2, it seems that 58 domains became 51 domains (which would mean 7 domains were removed) (p27,line 30-43). This should be explained/clarified.
--	---

REVIEWER	Dr. Kaye Shelton Lamar University, United States
REVIEW RETURNED	29-Apr-2019

GENERAL COMMENTS	Be specific - for example, surveys were left open for a couple of months is too vague. The biggest criticism of a Delphi is sloppy execution, be precise when describing your steps. Because you used face-to-face meeting, I would call this a Modified Delphi Study.
--

VERSION 1 – AUTHOR RESPONSE

Reviewer: 1

Please see minor comments below-

Page 3, line 55- What is “observation”?

Clinical added to better define what is meant by observations of visual signs.

Page 4, line 23- Provide full form for the abbreviation IVIS

Page 5, line 5- Provide full form for the abbreviation NHS

Full form of abbreviations added.

It is surprising that only 1 ophthalmologist was involved in the initial phases of the Delphi process and no ophthalmologist was present at the consensus meeting. I would think that it is important that ophthalmologists were better represented given that they are the primary eyecare personnel to diagnose and treat visual impairment in stroke survivors. Please acknowledge this in the limitations. We have acknowledged the limited response to the Delphi process by Ophthalmologists within the limitations of the study. In the UK, ophthalmologists are not the primary eye care provider for diagnosis and management of stroke-related visual impairment. This is done by orthoptists. Thus the composition of the steering group was appropriate. However, we acknowledge this is not the case in other countries and have added this consideration to the discussion of limitations.

In the abstract and main text, instead of re-listing each of the 9 visual assessments from the screening for the full vision assessment, it will be easier for the reader if the authors say- 9 assessments + Binocular vision assessment + Quality of life questionnaires. It is unclear why non-eyecare personnel cannot also perform the binocular vision assessment and quality of life questionnaires if they are able to perform the 9 other tests. It is also unclear how much more information these 2 additional tests will actually provide in helping to treat vision problems in stroke survivors.

We feel it is important to retain both lists in full, as some readers may be interested in the full assessment and not the screening assessment. Although there are nine visual assessments, which are referred to in the same way, the actual tests and person testing will be different between the screening and full assessment, i.e. more detailed tests for the full assessment for which non-eye care clinicians are not trained such as quantitative perimetry, ocular motility torsion assessment, etc. This difference between the screening and full assessment is defined and further clarified in the results on page 5, lines 11-18.

Reviewer: 2

I do have to mention one point, which relates to this article as well as is just a general comment/suggestion. The term "visual neglect/inattention". I would politely like to make a point to this terminology because I think this word confuses the reader. "Visual neglect/inattention" is the same thing so why put both words? Before my dissertation defense, I looked up the definition of "neglect" and then "inattention" and both are defined: "to pay little or no attention to". I think using multiple words to define this disorder is a HUGE problem. I advocate for the word "spatial neglect" as it is all encompassing. Then after stating that the person has spatial neglect then talk about the type of spatial neglect that the person has. So in this case, I would say that it is affecting the "visual domain". So many researchers and clinicians use different terminology to describe this disorder and this is one reason why I think spatial neglect is under-detected. Also, I have to comment that in this article, 2 of the outcomes that found from the reviews, for "assessing visual neglect" was room environment description and clock drawing. I would suggest these are measuring a different type of spatial neglect. I.e: the room description (from memory) tests representational neglect rather than spatial neglect of the vision domain. Seminal article: <https://www.ncbi.nlm.nih.gov/pubmed/16295118/>

Also, is the clock drawing also an assessment the visual domain? See here: <https://www.ncbi.nlm.nih.gov/pmc/articles/PMC3371137/>

Duplication of terminology simplified to visual neglect throughout.

Reviewer: 3

1. Page 7, line 18 – The authors state that “combination decisions” were made by the steering group. My understanding of a “combination decision” is that it a term used to describe when an individual makes a decision after listening/integrating views of others. Did one person of the steering group make the decision or was it achieved through another mechanism? And why is “decisions” plural?

There was one combination decision. Clarification has been provided that this was made by the steering group through discussion of the various test choices.

Specifically, COS do not address how outcomes (or in this case tests) should be defined or measured, but there is good guidance on how this next stage should be achieved; e.g. Prinsen CAC, Vohra S, Rose MR, Boers M, Tugwell P, Clarke M, et al. How to select outcome measurement instruments for outcomes included in a “Core Outcome Set”—a practical guideline. *Trials* 2016; 17:449. <https://doi.org/10.1186/s13063-016-1555-2> PMID: 27618914

2. I was surprised that no neurologists/stroke advanced practitioners were involved in the survey. This may be due to variations in medical systems between different countries. Where I practice the doctor or advanced practitioner is often the only person who screens the patients for deficits (and then

requests help from occupational therapists or eye experts) – therefore their input, at least in identifying best screening outcome measures, may be of interest.

We agree and this is likely due to variation in medical systems. We have highlighted in the limitations that this study was based in UK and Ireland and further work is needed to seek expert opinion internationally. Stroke advanced practitioners were involved, e.g. stroke occupational therapists.

3. What was the logic behind separating results in phase 2 round 2 (page 8, 3rd paragraph) into 3 different stake holder group outcomes? This could lead to individual groups reinforcing their biases, is that part of the intent and if so how is that helpful for this method? Also, given that the steering group changed this strategy and provided one compilation of all responses in round 3 after judging the stakeholder group results to be “similar in terms of percentage spread across the responses of 1 to 9” (p8, lines 34-35), will the graphs be made available as supplementary material, so that the reader can assess the differences/percentage spread themselves?

This method has been used in other core outcome set development processes and enables different stakeholder perspectives to be considered by all participants before re-rating. We have added this explanation into the section. Providing the feedback from all stakeholder groups to all participants in round 2 has been shown to facilitate consensus.

We have not provided the graphs in supplementary material because of the volume of graphs that were generated in this process – a considerable number even for supplementary material.

CONCERNS WITH REGARDS TO CONTENT/CONCLUSIONS:

Some of the caveats with regards to using such a consensus process were not sufficiently addressed. Specifically:

1. Participants were asked to score the “importance” of each domain in the Delphi survey. I do not have access to what instructions they were given, but it seems to me that it would be important to pre-define the ultimate goal prior to starting such a survey. What makes a domain “important”? Is it the impact on patient quality of life? Is it the impact on patient ability to participate in rehabilitation for other stroke-related deficits? Financial considerations, or relative benefit-for-time-unit-spent doing the assessment / cost-effectiveness? It may be that no further instructions were given to the survey participants, but the fact that the determination of “importance” may vary depending on the survey participant’s values should be noted in the discussion.

Participants were informed of the goal of the survey prior to starting as the survey consisted of an opening page detailing the study and instructions for survey completion. This has been clarified in the methods along with a sample of text from the introductory page.

2. Similarly, were participants aware of the planned Consensus in/out definitions (as described in p8 lines20) when they were scoring the outcome measures? (and were these definitions pre-defined or determined after reviewing the results?)

Added into the methodology that the consensus in/out definitions were defined a priori and that participants were unable of these cut offs.

3. The authors correctly point out that the low sample number and high attrition rates are limitations to the study, but they should also include some critical comments about the risks of using the Delphi method. For example, if a majority of those sampled have a misconception about one of the items evaluated, this type of group consensus will only further perpetuate/reinforce the misconception. We have added discussion to the limitations section.

OTHER DETAILS:

1. JK is listed as an author but has no contributions specified (p2, lines3-7)

We apologise for this omission and have now included JK in the author contributions.

2. Page 7, line 9 - Reference 15 refers to the Delphi method itself and therefore probably was not intended to be referred to as one of the 7 systematic reviews used to compile a preliminary list of outcomes

Reference moved.

3. Page 8, lines 18-30 – the methods of round 2 were confusing to me. I ultimately concluded that this meant that each participant who completed round 1 received 2 results back to them, their original score AND the summary of the scores from their corresponding group. If this interpretation is correct, perhaps the sentence “the results were presented by these stakeholder groups” (line 22-23) could be

rephrased...I initially interpreted this as “the stake holder groups” gave a presentation to the clinicians patients (as opposed to this describing a way of sorting/organizing and then presenting data)
 This section has now been rephrased; all participants were given the summarised results of all stakeholder groups.

4. Page 8, line 49 – there appears to be an “of” missing in “to score the quality [of?] evidence for outcomes”
 Corrected.

5. The link to the COMET initiative website (<http://http://www.comet-initiative.org/studies/details/275?result=true>) did not work for me when I attempted to access it. I would recommend that the authors make the protocol available as supplementary material for readers.
 Weblink has been replaced.

6. I was confused by the numbers of domains during different phases of the Delphi survey. Specifically after Phase 2 (p10, lines 47-58) the authors state that 3 outcomes were excluded and that “four outcomes were combined” and “could be combined into one domain” (line 54-55). I interpret that as 4 outcomes were combined into 1, which means that 3 would be subtracted. Therefore a total of 6 outcomes were removed between Phase 2 and Phase 3 – yet in figure 2, it seems that 58 domains became 51 domains (which would mean 7 domains were removed) (p27,line 30-43). This should be explained/clarified.
 This was a typo error and has been corrected.

Reviewer: 4

Be specific - for example, surveys were left open for a couple of months is too vague. The biggest criticism of a Delphi is sloppy execution, be precise when describing your steps.
 The specific amount of time the surveys were left open has been added - 10 weeks. We have also added further detail of the number of domains across each round of the survey.

Because you used face-to-face meeting, I would call this a Modified Delphi Study.
 We followed a defined process for the development of core outcome sets which include all three phases of outcome identification, Delphi survey and consensus meeting. The Delphi survey was conducted in its own right, hence it has not been described as modified.

VERSION 2 – REVIEW

REVIEWER	Varshini Varadaraj Johns Hopkins Wilmer Eye Institute, USA
REVIEW RETURNED	08-Jul-2019

GENERAL COMMENTS	The comments I raised in my previous review have been addressed adequately. I have no further comments.
---

REVIEWER	Kimberly Hreha UTMB, USA
REVIEW RETURNED	06-Jul-2019

GENERAL COMMENTS	Great work with the revisions.
--------------------------------

REVIEWER	Ania Busza University of Rochester USA
REVIEW RETURNED	22-Jul-2019

GENERAL COMMENTS	The authors have addressed all of my concerns
---

	(on a side note - I was surprised that "too many graphs" was given as a reason not to include data in supplementary material - I would think that this day and age there is usually plenty of space for supplementary material? In general, I think maximal availability of data is important to strive for in research publications - but I do not think that this specific issue should keep what is otherwise an interesting paper from being published.
--	--